energy/evolution/behaviour

rural electrification, rural development, developing countries, communication networks, mobility networks

**Author for correspondence:**
Hadrien Salat
e-mail: hadrien.salat@gmail.com

# Analysing the impact of electrification on rural attractiveness in Senegal with mobile phone data

Hadrien Salat[1,2], Markus Schläpfer[2,3], Zbigniew Smoreda[1] and Stefania Rubrichi[1]

[1]Sociology and Economics of Networks and Services Department, Orange Innovation Research, 44 Avenue de la République, Châtillon 92320, France
[2]Future Cities Laboratory, Singapore-ETH Centre, ETH Zürich, 1 Create Way, CREATE Tower #06-01, Singapore 138602, Republic of Singapore
[3]School of Computer Science and Engineering, Nanyang Technological University, 50 Nanyang Avenue, Singapore 639798, Republic of Singapore

HS, 0000-0003-0958-9715; SR, 0000-0002-5769-8935

Reliable and affordable access to electricity has become one of the basic needs for humans and is, as such, at the top of the development agenda. It contributes to socio-economic development by transforming the whole spectrum of people's lives—food, education, healthcare. It spurs new economic opportunities, thus improving livelihoods. Using a comprehensive dataset of pseudonymized mobile phone records, we analyse the impact of electrification on attractiveness for rural areas in Senegal. We extract communication and mobility flows from call detail records and show that electrification is positively and specifically correlated with centrality measures within the communication network and with the volume of incoming visitors. This increased influence is however circumscribed to a limited spatial extent, creating a complex competition with nearby areas. Nevertheless, we found that the volume of visitors between any two sites could be well predicted from the level of electrification at the destination and the living standard at the origin. In view of these results, we discuss how to obtain the best outcomes from a rural electrification planning strategy. We determine that electrifying clusters of rural sites is a better solution than centralizing electricity supplies to maximize the development of specifically targeted sites.

# 1. Introduction

Providing reliable and affordable access to modern energy services is specified as one of the top social priorities in the

United Nations' 2015 Sustainable Development Goals [1]. Getting access to energy can potentially transform the whole spectrum of people's lives: stored energy can be used to improve healthcare services, to help pump water for domestic usage or irrigation, to access safe fuel for cooking and heating; it positively impacts education by providing better lighting and access to technological support and it facilitates income generating activities by increasing overall productivity. It is therefore recognized as a critical element to enable human beings to thrive and no one should fall short of it. Yet, according to the International Energy Agency (IEA), nearly 12% of humanity lacked any access to electricity supply in 2018, including about half of the sub-Saharan African population [2].

Concurrently, a symbiotic relationship exists between access to energy and the development of information and communication technologies (ICT). Such technologies are, in their own right, already playing an essential role in global development as a communication tool, a facilitator of connections between individuals, information, services and markets [3,4]. The penetration rate of mobile phones has dramatically increased in the last decades, creating an unprecedented demand for electricity supply to both charge phones and power communication towers. Despite the wide coverage of mobile phone networks, usage remains low in Sub-Saharan rural areas, constrained, among other factors, by limited access rates to electricity (about 26% in 2018, compared to 74% in urban areas [2]). Yet, powering entertainment appliances such as radios, television sets and mobile phones, all linked to knowledge and information access and sharing and to the possibility of connecting with their urban counterpart, comes second after lighting in rural households' electricity usage [5,6]. This holds promise for establishing deep ties between energy access and the underlying network effect.

Much of the debate in the literature about the impact of electrification programmes on development outcomes centres around metrics directly related to productivity levels, income and quality of life [7]. A small, but growing, body of studies with a focus on mobile phone usage is emerging. Kirubi et al. [8] have conducted a detailed case study of a community-based electric micro-grid in rural Kenya and have found positive impact on Internet access at the community level. Lenz et al. [9] assess the impact of a large-scale rural electrification programme in Rwanda, connecting 370 000 new users within less than four years, and find significant increase in mobile phone ownership and usage. Armey et al. [10] assess the impact of access to electricity on digital divide in a panel of 40 low-income countries, finding an increase in within-countries adoption of the Internet. However, despite the overall positive impact on the uptake of the technology, very little has been written on the specificity of related usages, their spatial and temporal patterns, and the associated spillover and network effects. In a more recent work, Houngbonon et al. [11] consider usages and users' characteristics in analysing the effects of Senegalese rural areas electrification, finding a positive impact on digital inclusion and its potential economic benefits.

Building on these results, this paper seeks to investigate the contribution of energy access to the socio-economic development of rural areas in Senegal by proposing an original quantitative framework to measure the resulting gain in accessibility and attractiveness, accordingly we discuss the best planning strategies and eventually support progress in reaching a global target for universal access to energy services by 2030 [12]. We consider accessibility and attractiveness as alternative intermediate indicators of impact that are affected in the short run. They measure the opportunity for interaction and access to information and ideas, leading to potential increases in productivity [13]. Electrification indeed is expected to influence the attractiveness of concerned communities through the improvement of the efficiency of existing services and the establishment of new ones, and to make them more reachable and open to the flow of outside information and ideas [14]. We take advantage of a real-world dataset, gathered from mobile phone users in Senegal, complemented by census data and satellite images, to quantify the impact of rural electrification on the accessibility and attractiveness of rural areas. Measuring is central to the implementation of any global target. It is, on its own, a development agenda, especially in data-poor countries, where standard sources of data may not be available.

We use mobility and communication flow-based approaches. Antenna sites with higher centrality within the communication network generally detain more information and have greater control over arising opportunities [15–17]. Similarly, a higher diversity of contacts has been directly linked to higher incomes [18]. In parallel, the approximate residence location of mobile phone users can be estimated from mobile phone data, allowing us to map the mobility network of visitors from one area to another. Can we predict the volume and profile of visitor flows from the electrification rate of the destinations? We find that electrification is positively correlated with centrality measures within the communication network and with the volume of incoming visitors, which is distinct from other socio-economic indicators. This increased influence is however circumscribed to a limited spatial extent. We also find that the level of electrification at the destination and the living standard at the origin are good predictors for the volume of flows between any two sites. In view of these results, we discuss

the best planning strategy to electrify rural areas in Senegal. We determine that electrifying clusters of rural sites is a better solution than attempting to centralize electricity supplies to maximize the development of specifically targeted sites.

# 2. Material and methods

In this section, we provide details on four elements that are central to all subsequent analyses: the mobile phone dataset and related communication and mobility networks, the measurement of access to electricity from the census and satellite images, the creation of reference socio-economic indicators from the census, and the definition of 'rural' in our context. Any other data preparation or method that is specific to a subsection of the Results and discussion section is presented there.

## 2.1. Mobile phone data

Mobile phone networks are today one of the major vectors of information flows. A large number of services and activities in our everyday life pass through mobile phones, one of the most pervasive and ubiquitous technologies in the last decades. Mobile phone network operators collect a large amount of data on communications passing through their network infrastructure: date, duration, location, nature of communication (voice calls, SMS, applications, etc.). These data, namely Call Detail Records (CDRs), constitute an extremely rich source of information and, under well-specified ethical and data privacy regulations, can be analysed and exploited to answer very diverse and challenging questions.

Indeed, mobile phone data analysis has become today a mature research field with a wide range of applications [19], which has been further fostered by the Orange Data for Development Challenge [20]. The past few years have seen the rise of research studies that have shown how to turn these data into actionable information to improve social and operational efficiency. Areas of application include public health to minimize the spread of infectious diseases [21–23], national statistics to help mapping populations [24] and associated socio-economic indicators [18,25], the estimation of transportation and electricity demands [26–28] or the monitoring of population displacement after disasters [29].

For the purposes of this research, we use a pseudonymized dataset of mobile phone CDRs collected by Orange-Sonatel, the market leader, between January and December 2013 in Senegal. It consists of spatio-temporal information about the telecommunication activity: each record contains details about the caller's and the callee's IDs, timestamp, duration and type of communication, as well as an identifier of the antenna that handled the communication.

From the CDRs, we estimate the cell sites corresponding to the users' residences (i.e. their 'home sites') by following a standard approach that evaluates users' activity at night time (9 p.m. to 6 a.m.), the period of the day people are most likely to be at home [30]. We define the 'home site' of a user as the antenna site where they make and receive calls or text messages (SMS) the most. After removing holiday-periods from the dataset, we count, for each user and each distinct day of the year, the number of calls and text messages made and received at night by cell site, and select the site with the highest number of records associated. The final home site is the one with the highest number of selected days over the observed period. We then build a communication network and a mobility network. Nodes in both networks represent Orange–Sonatel cell sites located across mainland Senegal. Links are weighted, respectively, by the daily volume of exchanged calls between residents and by the average number of residents from one node visiting another node in a day. Both networks are directed (calls and trips have identified origins and destinations).

## 2.2. Access to electricity

We compute the electrification rate within each commune of Senegal from the information contained in the census conducted in 2013 by the Senegalese National Institute of Statistics (ANSD) [31]. The commune level is the smallest sub-division available for which a GIS shapefile is available. We have access to an extract with the answers of a sample of about 10% of each municipality's households, totalling around 1.25 M individuals. Among the questions asked, one relative to the source of lighting informs us whether individuals have access to a stable source of electricity (main power grid or solar systems), to an unreliable source (diesel engines) or no access at all. In this context, diesel engines refer to small individual units that are shared by a varying (small) number of households and do not

offer the same level of stability as solar systems, which benefit from high solar irradiation across the country and very low maintenance costs [5,32]. The electrification rate is taken as the ratio between the number of individuals with access to a stable source of electricity (main power grid or solar systems) and the total population count. We then compute the electrification rate around each antenna site by intersecting its Voronoi neighbourhood with the communes. The intersection is weighted by the population count, assuming a homogeneous distribution of the population inside each commune. This assumption, also made for the Grid Population of the World (GDW), albeit at a broader administrative scale (45 departments in contrast to 552 communes for Senegal), is reasonable, since the boundaries of all big and medium towns are known and identified as separate communes. As a control, we have computed the average night-time lights intensity inside each Voronoi cell from NOAA's open access 2013 satellite images [33]. The results were found in good agreement, and only those with the electrification rates based on the census data are shown.

## 2.3. Socio-economic indicators

To evaluate the socio-economic status of the population around each antenna site, we build four indicators based on the information available in the 2013 census. Each indicator is computed at commune level first, then at site level by weighted intersection with the mobile phone Voronoi cells, in the same manner used for electrification rates. A 'quality of accommodation' indicator is provided by the first component of a multiple correspondence analysis applied to the questions relative to the characteristics of the accommodation (type of house, tenure, main construction material, roofing, flooring, type of toilet, access to water, cooking appliances, waste disposal and waste water disposal). To make this indicator more independent from electricity, we do not use any of the questions involving ownership of electrical appliances and equipment. A second 'level of education' indicator is defined as a direct count of the average number of years spent studying either at school or in higher education. A third 'employment status' indicator is created by applying a principal correspondence analysis to the number of individuals in each employment category (inactive, unemployed, student, intern, fixed-term employed, permanently employed, self-employed, employer). A fourth 'combined socio-economic' indicator is given by the first component of a principal component analysis of the first three indicators (this component explains 93.08% of the variance of the three other indicators).

## 2.4. Rural classification

In the lack of a generally agreed definition of 'rural', we adopt the traditional approach consisting of defining 'urban' first, then defining rural sites as those that are not urban. This approach is used, for example, by the Food and Agriculture Organization (FAO) of the United Nations and by the United Nation Statistics Division (UNSD) [34]. It is admittedly not a perfect solution, since there is no consensual definition of 'urban' either. We choose a custom double criteria for this definition. Specifically, we classify any antenna site as 'urban' either if it falls within one of the communes tagged by the ANSD as a *commune d'arrondissement* (division into districts of the five biggest cities) or as a *commune urbaine* (urban communes) in their official December 2013 redefinition of administrative boundaries; or if the antenna site's Voronoi neighbourhood contains more than 1000 inhabitants per km². Although arbitrary, this threshold is commonly used, for example, by the FAO and by the Global Rural Urban Mapping Project (GRUMP) [34]. To evaluate the population density within each Voronoi cell, we rely on the official census. We do not use the land use corrected Landscan mapping to avoid introducing a bias with the electrification variable, since this method uses night-time lights to make the estimations. According to our classification, we count 708 'purely rural' sites and 879 'urban' sites out of the original 1587 sites.

## 2.5. Data availability

The 2013 Senegalese census data (10% sample of the full census) and the administrative boundary definitions can be accessed through the official website of the ANSD [31]. The night-time lights data can be accessed through NOAA's open database [33]. The detailed mobile phone data is proprietary and confidential. We obtained access to the mobile phone data from Sonatel within the framework of a research contract and with the authorization of the CDP (Senegalese data protection authority). For legal reasons, the full dataset cannot be published. However, access to the data can be requested from Sonatel on a contractual basis (contact: Département Produits Digitaux, Sonatel, Aissatou.Gningue@

**Table 1.** Spearman rank correlations between network measures and electrification rates.

| correlation | all sites | | rural sites | | rural sub-network | |
|---|---|---|---|---|---|---|
| | direct | partial | direct | partial | direct | partial |
| betweenness | 0.50* | 0.31* | 0.47* | 0.21* | 0.47* | 0.21* |
| leverage | 0.62* | 0.34* | 0.58* | 0.15* | 0.55* | 0.17* |
| social div. | 0.54* | −0.05 | 0.09* | −0.04 | −0.05 | 0.02 |
| spatial div. | 0.46* | 0.38* | 0.56* | 0.06 | 0.54* | 0.15* |
| distance | −0.78* | −0.21* | −0.34* | 0.04 | −0.23* | 0.04 |
| in/Out | −0.45* | −0.25* | −0.26* | −0.29* | −0.16* | −0.25* |

*$p$-value $< 0.05$.

orange-sonatel.com or post mail: Orange-Sonatel, 46 Boulevard de la République, BP 69 Dakar, Senegal). Mobile phone data aggregated at antenna site level, census information at Voronoi level and $R$ codes are available inside an Open Science Framework repository [35]. These are enough to recover all the results presented in the article.

# 3. Results and discussion

We first analyse the correlation between electrification and the importance of antenna sites, in particular rural sites, within the communication network. A site that is more central in the network, or that has more diverse contacts, usually detains more information and has greater control over arising opportunities [15–17]. It may also be perceived as more socially valuable by its neighbours and attracts more assets favourable to its development [36]. Such assets could be wealthier or more skilled visitors. To explore this aspect, we measure the size and profile of the flows of visitors in the second part of the section. In particular, we show that it is possible to predict satisfactorily the volume of flows between antenna sites from socio-economic and electrification indicators alone.

## 3.1. Electrification and importance within the communication network

The communication network based on all calls is used. Each node represents an antenna site. Each edge has two attributes: the average number of calls per day and the physical straight-line distance between the two antenna sites it is linking. For each node, we compute its unweighted and undirected betweenness centrality [37] (i.e. the number of shortest paths that pass through the node, independently of the direction travelled by the call) and its leverage centrality (i.e. the degree of the node compared to the degree of its neighbours). Since the betweenness centrality is used to measure how much information a node controls, only significant edges with a number of average calls greater than one per day were kept for this calculation. In addition, we compute the social diversity index and an ad hoc adaptation of the spatial diversity index [17]. The social diversity index measures how diverse the contacts of a node are by calculating the Shannon's entropy corresponding to the distribution of its calls among its contacts (including both incoming and outgoing calls). Eagle *et al.* [17] have shown that it is well correlated with income for a CDR dataset collected in the UK. Our spatial diversity index is defined similarly to the social diversity index, with the only addition of distance multiplied to the volume of calls between the node and each of its neighbours. Finally, we compute the average distance between each node and its neighbours, weighted by the volume of calls, and the ratio of in-calls to out-calls.

The Spearman rank correlations between each of these six indices and the average electrification rate around each tower are shown in table 1. The first two columns use the entire network, the next two columns use the values for rural sites only, while the last two columns use a recomputation of the seven indices over the sub-network containing only direct links between rural nodes. Since the electrification rate variable is highly correlated to the combined socio-economic indicator defined above (0.90 nationwide and 0.88 for rural areas), we additionally show in the table the partial correlations (i.e. the correlations, here in the sense of Spearman, between the residuals of two

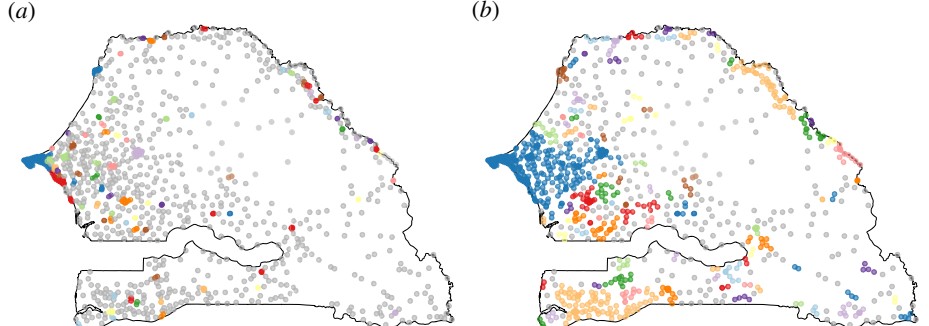

**Figure 1.** Clusters identifying all sites within a given distance of each other. (*a*) The distance is set to 5 km, resulting into 768 clusters. (*b*) The distance is set to 10 km, resulting into 364 clusters. When a cluster contains only one point, it is coloured in grey, otherwise a random colour is assigned to it.

variables when linearly predicted from a third control variable) between the network measures and the electrification rates, controlled by the combined socio-economic variable. This allows us to distinguish the specific contribution of electrification and show that the gain in centrality is not a coincidental consequence of a general perception of wealth, concomitant with electrification, triggering more calls with wealthier areas. The full *p*-values corresponding to all these correlations are provided in electronic supplementary material, table S1.

We can observe that all six indices are significant when the entire network and direct correlations are considered. We find that electrified areas are indeed more central within the network for both measures (betweenness and leverage). Their contacts are also more diverse and their inhabitants tend to have a higher out/in ratio than average. The highly negative correlation with the average distance between callers and callees, less pronounced for rural areas, can be explained by the high density of antennas within urban areas. Although slightly lower in value, all indices, with the notable exception of social diversity, are also statistically significant for rural sites. These values, although low in absolute terms, are quite significant in the context of partial correlations where other explanatory factors have been reduced. When considering the partial correlations, even though the correlations for the centrality measures are lower, the same conclusions remain. There appears to be therefore a (small) impact of electrification that is distinct from other living standard indicators. The ratio of in-calls to out-calls remains steadily negative in all cases, which shows that electrification is correlated with people initiating more calls with their contacts. By contrast, the social diversity, and in some cases the spatial diversity, as well as the average distance for rural areas, are all close to zero, which indicates that they are not *a priori* correlated specifically to electricity. This may result from the heavy constraints on mobility in rural areas not allowing to develop and maintain a diverse network beyond near geographical proximity. Interestingly, this could mean that, independently of electrification, the richer the people are, the more local their communication becomes.

Note that the high density of antennas within urban areas 'dilutes' their activity: although cities are expected to be highly central when considered as single entities, the importance of any particular node within the city is diminished by the competition from similar nodes near them. To compensate for this effect, we merge all antennas within a set number of kilometres of each other into one 'super-antenna'. For example, with a distance set to 5 km, this process allows us to aggregate all the major cities into single entities (figure 1*a*), and with a distance set to 10 km, we can isolate major continuous regions in terms of antenna density (figure 1*b*). According to our definition of 'rural', the full network contains 708 rural antenna sites out of 1587 sites, the 5 km aggregated network contains 515 purely rural clusters out of 768 clusters, and the 10 km aggregated network contains 235 purely rural clusters out of 364 clusters. With these aggregations, we can compare the structure of long-range calls with the structure of short-range calls.

The evolution of the Spearman correlations for all cases reported in table 1 is shown in figure 2 for aggregation distances comprised between 0 km and 15 km. For direct correlations, we observe that the aggregation process induces a steady decrease of the correlations with the centrality measures, which is much sharper for rural areas. The fact that the correlations with the social and spatial diversities rapidly become negative, except for the rural sub-network, indicates that the initial high diversity of calls is primarily due to intra-urban and urban to close-by rural calls, showing a clear divide of usage between urban and rural areas (see [38], reporting similar results). For partial

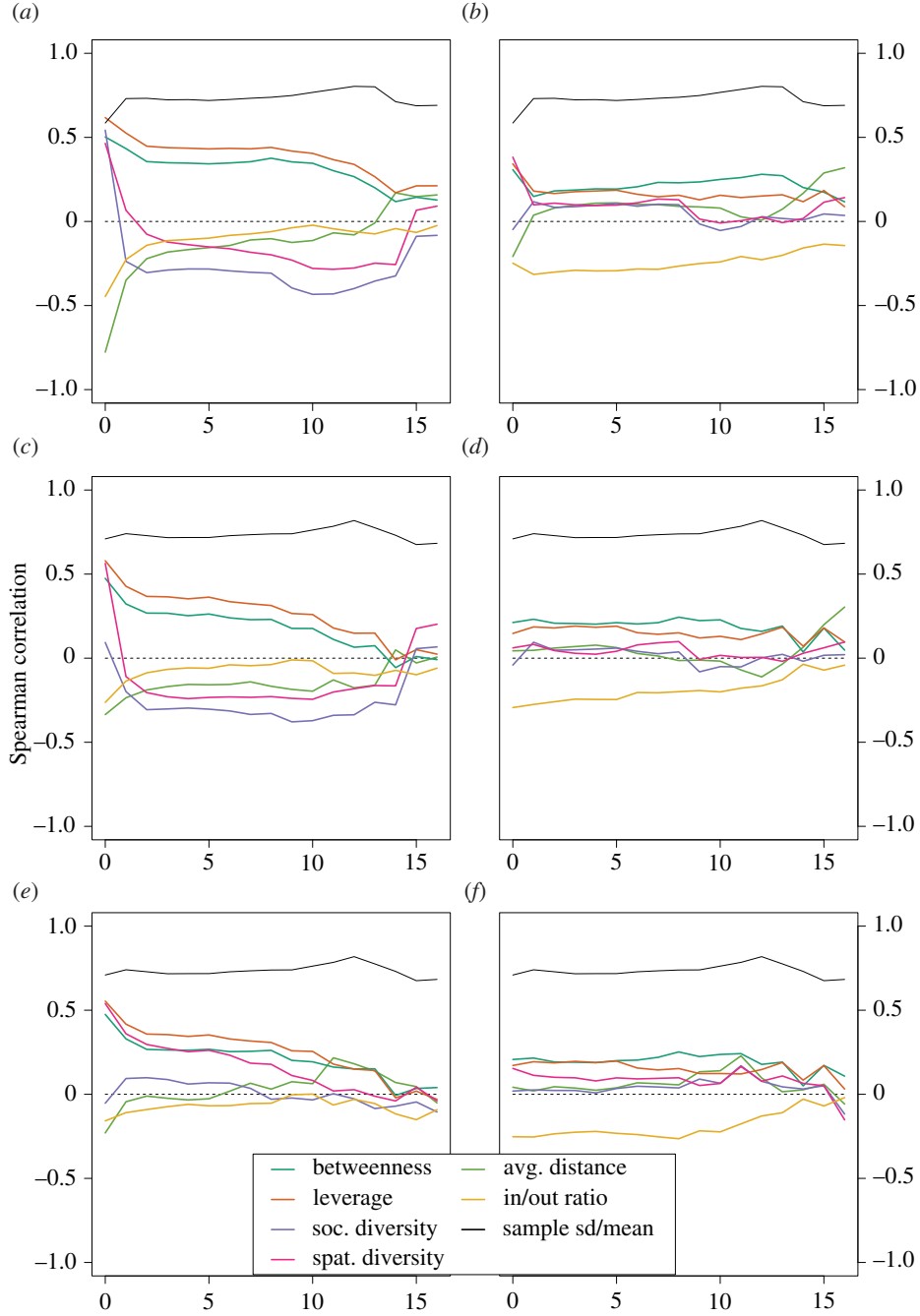

**Figure 2.** Spearman correlations through successive levels of aggregation. All antenna sites: (*a*) direct correlation and (*b*) partial correlation. Rural antenna sites: (*c*) direct correlation and (*d*) partial correlations. Rural sub-network: (*e*) direct correlation and (*f*) partial correlation. The *x*-axis indicates the merging threshold.

correlations, the lines appear more constant, although they seem to become unstable from around 6–8 km. All these observations indicate that the gain in importance linked to electrification is mainly local. To discard a possible smoothing of the electrification rate distributions within clusters with higher aggregation levels as a possible explanation for the convergence to zero of the correlations, we indicate with a black line the evolution of the ratio between the standard deviation and the mean of the distribution of electrification rates within the considered areas (all or only rural), and find that these lines are indeed almost constant. For reference, the full histograms of electrification rates within rural and urban areas are shown in electronic supplementary material, figure S1.

In summary, we have evidenced that electrification has a rather small correlation, yet positive and distinct from the correlation of other socio-economic indicators alone, on the centrality of both urban

**Figure 3.** Socio-economic status of visitors against level of electrification of visited location. Blue represents the level of education of visitors in mean number of years spent studying, green represents employment status, red represents the quality of the accommodation. The $x$-axis is the electrification rate as a ratio of people inside the area having access to electricity. All $p$-values for the black fittings (used only to quantify the upward trend) are smaller than $2.2 \times 10^{-16}$.

and rural inhabitants. This potential impact is quite local, especially in rural areas. In addition, there seems to be a noticeable divide between rural and urban areas. The higher social diversity for electrified rural areas close to urban areas indicates, however, that more interactions between rural and urban areas can become possible. In particular, the fact that electrification allows a higher outgoing to incoming calls is a positive sign that it can help initiate more interactions.

## 3.2. Visitor flows to electrified areas

Despite calls being immaterial, we have seen that the composition of the physical neighbourhood surrounding a node has a large influence on its characteristics within the network. We will now see that this is also true for the volume and socio-economic profile of visitors to rural antenna sites. We first study both the size and the socio-economic profile of the incoming visitor flows as a function of electrification. We then try to predict the volume of these flows using gravitation models from the factors that we found predominant: living standard at origin and electrification at destination.

### 3.2.1. Socio-economic profile of the visitors

A visitor is defined as an individual who made a call through an antenna site that is not their assigned home site. They inherit the average socio-economic profile of their home antenna site. In figure 3, the average quality of accommodation, level of education and employment status socio-economic indicators of each node's visitors is plotted against the node's level of electrification. We observe a clear correlation between electrification rate and average socio-economic status of visitors. To avoid any bias, we apply the same method as in §3.1 and compute the partial Spearman correlations between average quality of accommodation, level of education and status of employment of visitors on one side, and electrification rate of the visited location on the other side, controlled by the combined socio-economic indicator of the visited location. We find correlations of 0.27, −0.25 and −0.26, respectively, with $p$-values ranging from $e^{-28}$ to $e^{-25}$, for all areas, and correlations of 0.18, −0.27 and −0.24, with $p$-values ranging from $e^{-14}$ to $e^{-7}$ for rural areas among all areas. Interestingly, this means that electrification attracts visitors with higher living standards, but not visitors with higher levels of skills. This might be due to a dynamic where individuals from deprived areas and with higher education are more likely to have studied and worked away from their birth home, creating a bias in their visits towards less electrified areas. Meanwhile, the positive correlation between electrification and quality of accommodation of visitors could simply be due to the fact that the area around an antenna site is more likely to get electrified if it is already part of a larger wealthier area, or close to an already electrified city. Since frequent visitors are more likely to come from nearby areas, the apparent extra wealth of visitors could only be due to wealthier residents already living nearby before electrification.

To study the importance of the composition of the surrounding area, we invert the problem: how much more likely is it that a given visitor originating from a specific antenna site would choose a more electrified destination among the antenna sites in their neighbourhood? The mean distance

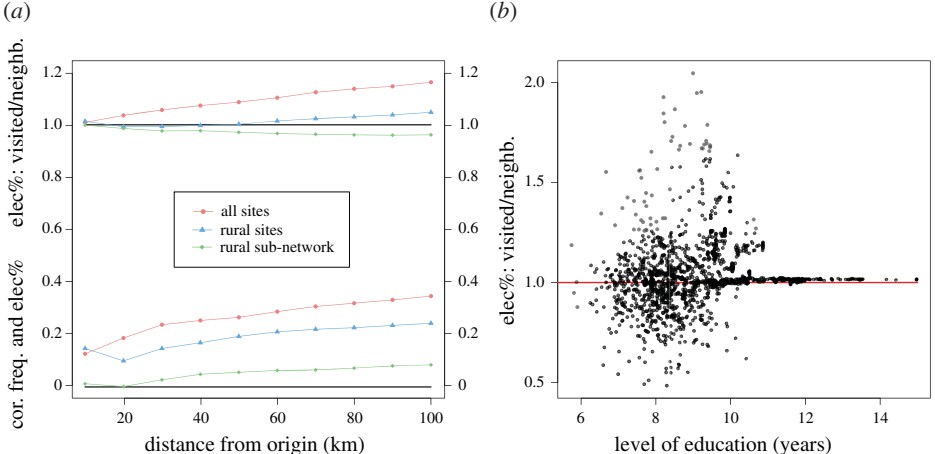

**Figure 4.** Impact of electrification at destination and socio-economic status at origin on the choice of a destination. (*a*) Ratio between the electrification rate of visited places and the electrification of a neighbourhood around the origin of travel and correlation between the frequency of visits and the electrification rate of visited places inside a neighbourhood around the origin of travel. Both are plotted against the size of the neighbourhood. (*b*) Ratio between the electrification rate of visited places and the electrification of a neighbourhood around the origin of travel against the level of education of the visitor.

travelled by a visitor between two antenna sites is about 37 km nationwide, 41 km between two rural sites and about 75 km if one end of the trip is a rural site and the other end is an urban site. Long trips are rare and usually involve travelling to and from the capital city, Dakar, or to and from the big mosque of Touba; they are difficult to complete within rural areas, due to the use of either donkey carts or buses as main means of transportation. To ensure that we encompass most types of travel, we define neighbourhoods of increasing sizes by 10 km increments, from 10 to 100 km. For each neighbourhood size, we compute the average electrification rate of visited areas (weighted by frequency of visits) divided by the average electrification rate of the neighbourhood (taking into account the population count around each antenna site). We also compute the direct Spearman rank correlation between the frequency of visits and the electrification rate of the destination. These results are reported in figure 4*a*. It is not possible to compute reliably the partial correlation controlled by the combined socio-economic indicator in this case, as the samples inside each neighbourhood are too small to smooth the added variance. We observe that more electrified areas have an edge, in the sense that they are more likely to be visited, over less electrified areas nationwide, although it is less pronounced for rural areas. However, when we only consider trips in between rural areas, electrification appears to be at a slight disadvantage for the average electrification rate indicator and only minor advantage for the non-partial correlation indicator. This can be explained by the competition that results from the urban areas that are often found near well-electrified villages: fellow rural visitors are more likely to bypass the other villages and go directly to the urban areas.

Furthermore, wealthier visitors do not seem to be influenced by the electrification rate of the destination they choose. This can be seen in figure 4*b*, where the average level of education of a visitor is plotted against the ratio of electrification rates between the visited locations and all sites within a 20 km radius (some very similar plots for employment status and quality of accommodation can be found in electronic supplementary material, figure S2). The socio-economic status of the visitors does have an impact, however, on their probability of starting a journey (we find a Pearson correlation of 0.63–0.47 for rural areas—between the combined socio-economic indicator at origin and the size of outgoing flows). Similarly, the electrification rate of the destination is very well correlated to the size of the overall incoming flows (Pearson coefficient of 0.68–0.43 for rural areas). In conclusion, if a site is electrified, it has a higher chance to be picked as a destination, but it will be chosen as a destination by residents of the surrounding area regardless of their own level of wealth (although wealthier residents do travel more). This confirms that the correlations identified in figure 3 are more due to the pre-existing composition of the neighbourhood, than they are a consequence of electrification triggering visits from individuals with a higher profile on average. In other words, electrification gives a competitive advantage in terms of volume of visits, but not in terms of visitor profiles.

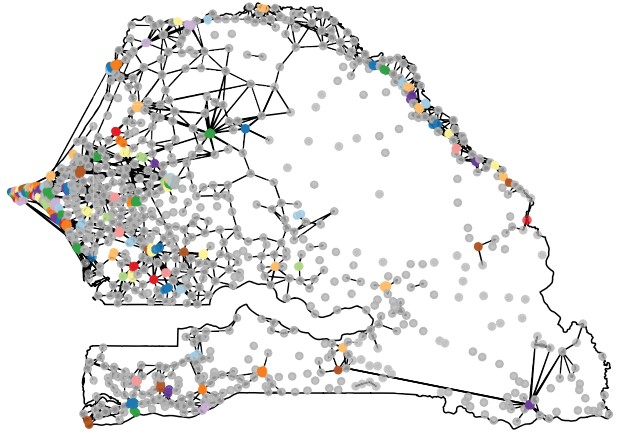

**Figure 5.** Flows of visitors between antenna sites. Only the flows containing an average number of daily visitors above 50 are drawn. When a cluster contains only one point, it is coloured in grey, otherwise a random colour is assigned to it.

### 3.2.2. Prediction of the volume of visits

Since we have observed that the volume of visits between any two sites depends heavily on the socio-economic status at origin and the electrification at destination, we can try to predict the flows directly from these variables using gravitation models [39]. The general form of the gravitation model is

$$T = k\frac{O_1^{\alpha_1} \cdots O_p^{\alpha_p} D_1^{\beta_1} \cdots D_q^{\beta_q}}{d^\gamma},$$
(3.1)

where $T$ is the size of the flows between an origin and a destination, $\{O_i\}_{1 \leq i \leq p}$ and $\{D_j\}_{1 \leq j \leq q}$ are a set of variables depending, respectively, on the origin and the destination, $d$ is the distance between the origin and the destination, and $k$, $\{\alpha_i\}_{1 \leq i \leq p}$, $\{\beta_j\}_{1 \leq j \leq q}$ and $\gamma$ are parameters to be determined. Often, the variables at origin and at destination are reduced to a single variable, the size of the population, but we can consider any variable that may have predictive power, such as electrification rate and level of wealth. In particular, we can put directly in competition several variables to identify which are the most important factors in driving the volume of the flows.

This kind of modelling is sensitive to the modifiable areal unit problem [40]. In our case, we are trying to model simultaneously intra-urban short-range flows and long-range flows between far apart antenna sites. A single value of the distance parameter $\gamma$ may not be enough to characterize these very different situations. Rather than merging according to the arbitrary administrative boundaries, we tackle this issue by applying a hierarchical clustering algorithm to the matrix containing the distances between all our antenna sites. We then merge all the branches of the resulting dendrogram whose distance is less than 5 km. This results in the 817 clusters visible in figure 5. Out of these, 523 clusters are considered purely rural under our definition of rural. Although in appearance similar to the merging method used in §3.1, this method produces a different output. In the dendrogram, the branches are considered situated at the barycentre of the leaves they contain. As a result, the clusters contain only a few antenna sites that are closer to each other than to the rest of the sites, rather than a continuous stretch of nearby sites. In addition, we use night-time lights intensity instead of electrification rates for this particular modelling, since the night-time lights intensity is smoother and therefore more stable under an exponent.

The parameters $\{\alpha_i\}_{1 \leq i \leq p}$, $\{\beta_j\}_{1 \leq j \leq q}$ and $\gamma$ obtained by multilinear regression of the logs for different choices of variables at origin and destination are shown in table 2. In each case, the quality of the prediction is shown in the first column as a Pearson $r^2$, and the parameter exponents assigned to each predictive variable are shown next. We observe that both night-time lights intensity and socio-economic status at destination are better predictors than population size when all trips are considered. Similarly, socio-economic status at origin is a better predictor than population at origin. We can achieve significantly better results when population, socio-economic status at origin and night-time lights intensity at destination are considered together. When origin constraints are added, i.e. when we force the total outgoing volume of visitors from each cluster to be equal to its known value [39], we can achieve a $r^2$ of 0.91 for the trips between all clusters. The same method applied within the

**Table 2.** Gravitation models: quality of predictions and fitted parameters. Pop stands for population, Ele for night-time lights intensity, Acc for quality of accommodation, Edu for level of education, Emp for employment status and Com for combined socio-economic index. The last model in the bottom row is an origin constrained variant of the one immediately above. The $p$-values for all exponents are $<2.2 \times 10^{-16}$.

| O/D | $r^2$ | $\alpha$ | $\beta$ | $\gamma$ |
|---|---|---|---|---|
| Pop/Pop | 0.59 | 0.96 | 0.79 | 1.26 |
| Pop/Ele | 0.84 | 0.88 | 0.59 | 1.14 |
| Pop/Acc | 0.79 | 0.89 | 2.56 | 1.21 |
| Pop/Edu | 0.72 | 0.96 | 6.34 | 1.31 |
| Pop/Emp | 0.74 | 0.96 | 2.28 | 1.32 |
| Pop/Com | 0.78 | 0.91 | 2.33 | 1.24 |
| Pop/Acc + Ele | 0.84 | 0.87 | 0.48 + 0.49 | 1.14 |
| Acc/Pop | 0.68 | 2.67 | 0.75 | 1.26 |
| Edu/Pop | 0.54 | 6.23 | 0.85 | 1.37 |
| Emp/Pop | 0.61 | 2.35 | 0.83 | 1.37 |
| Com/Pop | 0.63 | 2.39 | 0.78 | 1.30 |
| Pop + Acc/Pop + Ele | 0.87 | 0.70 + 0.81 | 0.26 + 0.45 | 1.07 |
| Above + constraints | 0.91 | n.a. | 0.29 + 0.46 | 1.11 |

rural sub-network achieves a $r^2$ of 0.65 with coefficients $\beta_1 = 0.18$ for population, $\beta_2 = 0.10$ for electrification and $\gamma = 2.28$ for distance.

The exponent for distance, $\gamma$, is quite similar for all cases at national level and is relatively close to 1, which indicates a relatively homogeneous slightly super-linear dependency on distance. This can be compared to the exponent $\gamma = 2.28$ for the rural sub-network, indicating a much higher impact of the direct cost of travel in rural areas. Recent work, which includes data from the Dakar region, shows that if distance is taken into consideration together with frequency of visitation, this exponent should be close to 2 [41]. Similarly, the size of the population remains a sub-linear factor with a relatively homogeneous exponent in all cases (around 0.9 when considered at the origin and 0.8 when considered at the destination), while the exponent for education is over 6, much higher than those for the other socio-economic indicators (between 2 and 3), and for electricity (around 0.6). The relative strengths of these exponents reflect their respective impact on the volume of predicted flows, and therefore reflect the impact of their associated variable on mobility. The exponent of over six associated with the level of education indicates that each additional year of study favours significantly the tendency to travel. However, the variables only represent their strict definition, and general quantitative conclusions cannot be drawn on the relative 'power' as a driver for mobility of employment compared to education from these exponents.

The results confirm those obtained in the previous section. Socio-economic status at origin and electrification at destination are good predictors for the volume of flows. In addition, we can observe that electrification has an impact that is independent of other socio-economic characteristics, since adding night-time lights intensity at destination to quality of accommodation at destination slightly increases the predictive power compared to quality of accommodation at destination alone.

## 4. Conclusion

Throughout the previous section, we have shown that electrification appears to bring some positive outcomes, which are distinct from other indicators of socio-economic status. However, these outcomes are usually small and, more importantly, hinge on the pre-existing composition of the larger surrounding area. Most of the trips between antenna sites cover less than 50 km. This creates an important regional effect. In §3.1, we have identified that electrification marginally increases the centrality of antenna sites, even in the rural sub-network. However, this increase disappears rapidly when sites are merged using a buffer of increasing size. In §3.2, we have shown that electrification

generally increases the volume of visitors. Nonetheless, the socio-economic profile of these visitors is determined by the socio-economic profile of the individuals already living nearby. Electrification does attract a larger number of wealthy visitors if these profiles already exist in the larger surrounding area, but does not draw wealthier visitors from further away. We have also found that people living in electrified areas are slightly more likely to initiate contact effectively with other antenna sites. This positive trend exists at all scales. By contrast, the higher the socio-economic profile of an individual is, the more local their communication gets. In particular, urban dwellers rarely engage contact with rural dwellers. Furthermore, it appears that the proximity of urban areas near electrified rural areas can be detrimental to these, as they are more likely to be bypassed by visitors from other rural areas.

How positive is it to electrify rural areas then? If such an area lies close to an urban area or lies inside a relatively wealthy region, it will benefit from a slight competitive edge compared to the non-electrified rural sites in the same area, and will potentially attract a larger number of wealthy visitors. By contrast, if such an area is more isolated, it is unlikely that it will draw wealthy visitors from afar, although it will still benefit from a slightly higher centrality among its local rural cluster, increasing its perceived importance in the region. Considering that isolated rural areas are generally quite poor and given the slowness of changes following electrification, a strategy based on the development of selected isolated rural areas by favouring the centralization of nearby opportunities does not appear efficient. On the contrary, since electrification is well correlated to all indicators of socio-economic status at the nation scale and since the development of an area seems largely determined by the wealth of its surroundings, our results support strategies that favour global solutions to electrify an entire cluster of sites, rather than a diversity of limited specific solutions.

Ethics. This research uses data from Senegal. It was approved by the Senegalese Commission de Protection des Données Personnelles (private data protection commission) on 13 July 2015 as part of the 'Data for Development (D4D)' project. This analysis only uses data that was pseudonymized by the mobile phone operator before accession by the authors.
Data accessibility. The 2013 Senegalese census data (10% sample of the full census) and the administrative boundary definitions can be accessed through the official website of the ANSD (http://www.ansd.sn/index.php?option= com_content&view=article&id=134&Itemid=262). The night-time lights data can be accessed through NOAA's open database (https://ngdc.noaa.gov/eog/dmsp/downloadV4composites.html). The detailed mobile phone data are proprietary and confidential. We obtained access to the mobile phone data from Sonatel within the framework of a research contract and with the authorization of the CDP (Senegalese data protection authority). For legal reasons, the full dataset cannot be published. However, access to the data can be requested from Sonatel on a contractual basis (contact: Departement Produits Digitaux, Sonatel, Aissatou.Gningue@orange-sonatel.com or post mail: Orange-Sonatel, 46 Boulevard de la Republique, BP 69 Dakar, Senegal). Mobile phone data aggregated at antenna site level, census information at Voronoi level and R codes are available inside an Open Science Framework repository (https://osf.io/mqzun/). These are enough to recover all the results presented in the article.
Authors' contributions. H.S., Z.S., M.S. and S.R. designed research; H.S. and S.R. performed research and wrote the paper; Z.S. curated the data.
Competing interests. We declare we have no competing interests.
Funding. H.S. was supported by Orange Innovation Research, through the Orange Labs-Sonatel-ETH Singapore SEC Research Agreement no. H11283 and a one-year postdoctoral fellowship. M.S. acknowledges the Future Cities Laboratory at the Singapore-ETH Centre, which was established collaboratively between ETH Zurich and Singapore's National Research Foundation (FI 370074016) under its Campus for Research Excellence and Technological Enterprise Programme.
Acknowledgements. We gratefully thank Georges Vivien Houngbonon and Erwan Le Quentrec for their helpful feedback on the study design and for the valuable insights into the field. We also thank Valerie Pla, director of the Orange research program 'Digital Emerging Countries', for supporting the research.

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
