## [Peer Review File · Royal Society Open Science]

Review History

RSOS-201898.R0 (Original submission)

Review form: Reviewer 1

Is the manuscript scientifically sound in its present form?

No

Are the interpretations and conclusions justified by the results?

No

Is the language acceptable?

Yes

Do you have any ethical concerns with this paper?

No

Have you any concerns about statistical analyses in this paper?

Yes

Recommendation?

Major revision is needed (please make suggestions in comments)

Comments to the Author(s)

The manuscript addresses the assessment of the level of electrification of sites on Senegal on its attractiveness for visitors which is a very relevant topic in the context of socio-economic development in African countries. The approach based on the analysis of call detail records performed in the manuscript is also interesting and suitable to generate novel results. However, in my opinion there are several points the authors should address before the manuscript can be accepted:

Why diesel engines are considered to be unreliable electric sources while solar systems are considered reliable? Is this because by diesel engines refer only to very small generators for a single household which may not be properly maintained? Also in what sense photovoltaic installations which are subject to weather variability are considered fully reliable? Is this because only large photovoltaic installations backed by storage systems (batteries) are considered?.

In the definition of the electrification rate, it is not clear what is understood by "individuals with a stable access", does this includes the individuals with households connected to "reliable sources" or does it includes also diesel engines?. Either way, authors built the index on the basis of a binary option (an individual has "stable access" or nothing), so it does not takes into account the existence of different reliability levels on the electrical access. A gradual index seems to me more appropriate, for instance assign 1 to individuals with access to reliable sources, 0.5 to those with access to less reliable sources and 0 to those with no access, and then take an average over the population.

In the first paragraph of 3(a), does the call distribution used to calculate the entropy for the social diversity index includes both incoming and outgoing calls or only outgoing?

In the paragraph before table 1, authors indicate that the last two columns of the table use a recomputation of the seven indices over the sub-network containing only rural nodes. However it is not explained how the recomputation is performed. For instance if in the full network A is a urban node and B and C are rural nodes, with calls from B to A and from A to C but no calls from B to C, the sub-network does have a link between B to C or not?

I suggest that at the beginning of the paragraph before table 1, "Spearman correlations" are refereed as "Spearman rank correlations".

In table 1 authors distinguish between direct and partial correlations, a distinction which is not evident for non-statisticians. Besides table 1, partial correlations are extensively used in the rest of section 3(a) as well as in 3(b). Therefore I suggest the authors to briefly explain, either here or better in sect. 2 materials and methods, what is meant by a partial correlation between two variables controlled by a third variable and how this is evaluated using Spearman rank correlations.

In the paragraph before table 1, it is not clear what the authors mean by of the "the electrification rate variable is highly correlated to our combined socio-economic indicator". Up to this point, authors have defined 6 indexes but there is no definition of a "combined socio-economic indicator". The "combined socio-economic indicator" is also used later in section 3 (b) (i), however neither there seems to be clearly defined. Is this indicator the sum of the level of education, the employment and the quality of accommodation?. If so, shouldn't the sum be weighted since education, according to fig. 4b ranges from 6 to 15 while the values of the quality of

accommodation, according to fig. S2, do not go beyond 3.1. A precise definition of the combined socio-economic indicator should be given in the main text.

As authors note in the paragraph after table 1, social diversity direct correlation with electrification is close zero in rural areas, however there is no discussion on this. I suggest to add a discussion on possible causes.

Spearman rank correlation values in between 0.2 and 0.4 are typically considered weak and those below 0.2 very weak. All partial correlations shown in table 1 and in Fig. 2 are below 0.4. I suggest authors to include this in the discussion and, if appropriate, put it in the context of the methodology used to evaluate the partial correlations.

In fig. 3 in section 3(b) (I) authors plot a straight lines on top of the data with an indication of the slope. However the data clearly does not have a linear behavior: It has a lower tail for low levels of electrification and an raising on for high levels of electrification typically associated to urban cells. In the text describing this figure, authors indicate high levels of correlation but the value of the correlation is not given and it is not clear that they refer to Spearman rank correlation or to Pearson linear correlation (used later for fig. 4). I think the lines associated to linear correlation should not be plotted and instead rely on Spearman rank correlation which is clearly more adequate.

In section 3 (b) (i) fig. 4, authors make use of the Pearson correlation between the frequency of visits and the electrification rate. Why Pearson correlation is used here instead of Spearman rank correlation?. Authors should justify why one should expect a linear relation between the quantities considered here, for instance by plotting, either in the text or the supplementary material, the frequency of visits versus the electrification rate.

In section 3 (b) (ii), authors consider several possible variables for the gravitation models, and the results for the adjusted parameters are shown in Table 2. Authors also include a discussion on the suitability of the different parameters which is fine. However, there is no much discussion on the values of the exponents. For instance, the exponent of the distance, γ , is quite similar for cases shown in table 2 and relatively close to 1. However α and β are quite different, for instance for Pop/Edu, the exponent of education, β is over 6 while for Pop/Ele β less than 0.6. I suggest the authors to discuss the values of the different exponents and in what sense these values can be considered as reflecting a physical mechanism for mobility. For instance, to which extend it make sense to consider the level of education to power 6?.

In the same section, for the case of considering the rural sub-network, I think that authors should discuss why the agreement is worst ($R^2=0.65$) than for the full network and to which extend this may imply that the mechanisms for mobility in urban and rural areas are different. Also the destination exponents β (0.18 and 0.1) are more close to zero than to 1, does this implies that the population and electrification of the destination play a little role in the mobility?. Besides, what are the values for α_1 and α_2 ?

Review form: Reviewer 2

Is the manuscript scientifically sound in its present form?

Yes

Are the interpretations and conclusions justified by the results?

Yes

Is the language acceptable?

Yes

Do you have any ethical concerns with this paper?

No

Have you any concerns about statistical analyses in this paper?

No

Recommendation?

Accept as is

Comments to the Author(s)

In my view, this is a very good piece of research. The authors perform a thorough, convincing analysis to investigate the relations between electrification and rural attractiveness in Senegal. To do so, a range of datasets are incorporated and performed a sound analysis.

As minor comments, I would recommend to adapt some of the choices of words, hinting that the analysis shows causal relations, and simply correlations.

Also, it appears from the map that most cities are located at the boundaries of the country, and I wondered if distance from boundary could be correlated to centrality measures, hence having an effect on the analysis.

Decision letter (RSOS-201898.R0)

Dear Dr Salat

Firstly, please accept our sincere apologies for the unusual delays incurred during peer-review of your manuscript. It initially proved extremely difficult to secure reviewers for your paper, and the current pandemic situation inevitably led to delays; particularly over the winter period.

The Editors assigned to your paper RSOS-201898 "Analysing the impact of electrification on rural attractiveness in Senegal with mobile phone data" have now received comments from reviewers and would like you to revise the paper in accordance with the reviewer comments and any comments from the Editors. Please note this decision does not guarantee eventual acceptance.

Please submit your revised manuscript and required files (see below) no later than 21 days from today's (ie 16-Jun-2021) date. Note: the ScholarOne system will 'lock' if submission of the revision is attempted 21 or more days after the deadline. If you do not think you will be able to meet this deadline please contact the editorial office immediately.

on behalf of Professor Weisi Guo (Associate Editor) and R. Kerry Rowe (Subject Editor)
openscience@royalsociety.org

Reviewer comments to Author:

Reviewer: 1

Comments to the Author(s)

The manuscript addresses the assessment of the level of electrification of sites on Senegal on its attractiveness for visitors which is a very relevant topic in the context of socio-economic development in African countries. The approach based on the analysis of call detail records performed in the manuscript is also interesting and suitable to generate novel results. However, in my opinion there are several points the authors should address before the manuscript can be accepted:

Why diesel engines are considered to be unreliable electric sources while solar systems are considered reliable? Is this because by diesel engines refer only to very small generators for a single household which may not be properly maintained? Also in what sense photovoltaic installations which are subject to weather variability are considered fully reliable? Is this because only large photovoltaic installations backed by storage systems (batteries) are considered?.

In the definition of the electrification rate, it is not clear what is understood by "individuals with a stable access", does this includes the individuals with households connected to "reliable sources" or does it includes also diesel engines?. Either way, authors built the index on the basis of a binary option (an individual has "stable access" or nothing), so it does not takes into account the existence of different reliability levels on the electrical access. A gradual index seems to me more appropriate, for instance assign 1 to individuals with access to reliable sources, 0.5 to those with access to less reliable sources and 0 to those with no access, and then take an average over the population.

In the first paragraph of 3(a), does the call distribution used to calculate the entropy for the social diversity index includes both incoming and outgoing calls or only outgoing?

In the paragraph before table 1, authors indicate that the last two columns of the table use a recomputation of the seven indices over the sub-network containing only rural nodes. However it is not explained how the recomputation is performed. For instance if in the full network A is a urban node and B and C are rural nodes, with calls from B to A and from A to C but no calls from B to C, the sub-network does have a link between B to C or not?

I suggest that at the beginning of the paragraph before table 1, "Spearman correlations" are refereed as "Spearman rank correlations".

In table 1 authors distinguish between direct and partial correlations, a distinction which is not evident for non-statisticians. Besides table 1, partial correlations are extensively used in the rest of section 3(a) as well as in 3(b). Therefore I suggest the authors to briefly explain, either here or better in sect. 2 materials and methods, what is meant by a partial correlation between two variables controlled by a third variable and how this is evaluated using Spearman rank correlations.

In the paragraph before table 1, it is not clear what the authors mean by of the "the electrification rate variable is highly correlated to our combined socio-economic indicator". Up to this point, authors have defined 6 indexes but there is no definition of a "combined socio-economic indicator". The "combined socio-economic indicator" is also used later in section 3 (b) (i), however neither there seems to be clearly defined. Is this indicator the sum of the level of education, the employment and the quality of accommodation?. If so, shouldn't the sum be weighted since education, according to fig. 4b ranges from 6 to 15 while the values of the quality of accommodation, according to fig. S2, do not go beyond 3.1. A precise definition of the combined socio-economic indicator should be given in the main text.

As authors note in the paragraph after table 1, social diversity direct correlation with electrification is close zero in rural areas, however there is no discussion on this. I suggest to add a discussion on possible causes.

Spearman rank correlation values in between 0.2 and 0.4 are typically considered weak and those below 0.2 very weak. All partial correlations shown in table 1 and in Fig. 2 are below 0.4. I suggest authors to include this in the discussion and, if appropriate, put it in the context of the methodology used to evaluate the partial correlations.

In fig. 3 in section 3(b) (I) authors plot a straight lines on top of the data with an indication of the slope. However the data clearly does not have a linear behavior: It has a lower tail for low levels of electrification and an raising on for high levels of electrification typically associated to urban cells. In the text describing this figure, authors indicate high levels of correlation but the value of the correlation is not given and it is not clear that they refer to Spearman rank correlation or to Pearson linear correlation (used later for fig. 4). I think the lines associated to linear correlation should not be plotted and instead rely on Spearman rank correlation which is clearly more adequate.

In section 3 (b) (i) fig. 4, authors make use of the Pearson correlation between the frequency of visits and the electrification rate. Why Pearson correlation is used here instead of Spearman rank correlation?. Authors should justify why one should expect a linear relation between the quantities considered here, for instance by plotting, either in the text or the supplementary material, the frequency of visits versus the electrification rate.

In section 3 (b) (ii), authors consider several possible variables for the gravitation models, and the results for the adjusted parameters are shown in Table 2. Authors also include a discussion on the suitability of the different parameters which is fine. However, there is no much discussion on the

values of the exponents. For instance, the exponent of the distance, γ , is quite similar for cases shown in table 2 and relatively close to 1. However α and β are quite different, for instance for Pop/Edu, the exponent of education, β is over 6 while for Pop/Ele β less than 0.6. I suggest the authors to discuss the values of the different exponents and in what sense these values can be considered as reflecting a physical mechanism for mobility. For instance, to which extent it make sense to consider the level of education to power 6?

In the same section, for the case of considering the rural sub-network, I think that authors should discuss why the agreement is worst ($r^2=0.65$) than for the full network and to which extent this may imply that the mechanisms for mobility in urban and rural areas are different. Also the destination exponents β (0.18 and 0.1) are more close to zero than to 1, does this implies that the population and electrification of the destination play a little role in the mobility?. Besides, what are the values for α_1 and α_2 ?

Reviewer: 2

Comments to the Author(s)

In my view, this is a very good piece of research. The authors perform a thorough, convincing analysis to investigate the relations between electrification and rural attractiveness in Senegal. To do so, a range of datasets are incorporated and performed a sound analysis.

As minor comments, I would recommend to adapt some of the choices of words, hinting that the analysis shows causal relations, and simply correlations.

Also, it appears from the map that most cities are located at the boundaries of the country, and I wondered if distance from boundary could be correlated to centrality measures, hence having an effect on the analysis.

===PREPARING YOUR MANUSCRIPT===

If you have been asked to revise the written English in your submission as a condition of publication, you must do so, and you are expected to provide evidence that you have received language editing support. The journal would prefer that you use a professional language editing service and provide a certificate of editing, but a signed letter from a colleague who is a native

speaker of English is acceptable. Note the journal has arranged a number of discounts for authors using professional language editing services (<https://royalsociety.org/journals/authors/benefits/language-editing/>).

===PREPARING YOUR REVISION IN SCHOLARONE===

<https://royalsociety.org/journals/authors/author-guidelines/#supplementary-material> to include a suitable title and informative caption. An example of appropriate titling and captioning may be found at https://figshare.com/articles/Table_S2_from_Is_there_a_trade-

off_between_peak_performance_and_performance_breadth_across_temperatures_for_aerobic_sc
ope_in_teleost_fishes_/3843624.

Author's Response to Decision Letter for (RSOS-201898.R0)

See Appendix A.

RSOS-201898.R1 (Revision)

Review form: Reviewer 1

Is the manuscript scientifically sound in its present form?

Yes

Are the interpretations and conclusions justified by the results?

Yes

Is the language acceptable?

Yes

Do you have any ethical concerns with this paper?

Yes

Have you any concerns about statistical analyses in this paper?

No

Recommendation?

Accept with minor revision (please list in comments)

Comments to the Author(s)

Authors have successfully addressed most of the points raised in my first report. However, in my opinion the point concerning the discussion on the exponents obtained for the gravity model has not been properly addressed. The answer of the authors seems to imply that the exponents obtained depend on the scale of the variables, and this should not be the case. In a gravity model the value as described by Eq. (3.1) the value of the constant k certainly depends on scale for the distance but the exponent γ should not. What (3.1) implies is that If, for instance, the distance is duplicated, then T decreases by a factor 2^{γ} no matter the scale in which distance has been measured (m, Km, as a fraction of the country size,...). Similarly exponents α_i and β_i should not depend on the units used to measure O_i and D_i , while the constant k does. This is why for a gravity law a discussion on the values of the exponents is relevant.

Therefore I consider that the text that authors have added at the end of 3 (b) (i) is completely misleading, besides the fact that has been added in subsection (i), before the introduction of the gravity law in subsection (ii).

As noted in my first report, I think the authors should include a discussion on the values of the exponents. For instance, the exponent of the distance, γ , is quite similar for cases shown in table 2 and relatively close to 1. However α and β are quite different, for instance for Pop/Edu, the exponent of education, β is over 6 while for Pop/Ele β less than 0.6. I suggest the authors to discuss the values of the different exponents and in what sense these values can be considered as reflecting a physical mechanism for mobility. For instance, to which extend it make sense to consider the level of education to power 6?.

Once this point has been addressed, in my opinion the manuscript can be accepted for publication.

Review form: Reviewer 2

Is the manuscript scientifically sound in its present form?

Yes

Are the interpretations and conclusions justified by the results?

Yes

Is the language acceptable?

Yes

Do you have any ethical concerns with this paper?

No

Have you any concerns about statistical analyses in this paper?

No

Recommendation?

Accept as is

Comments to the Author(s)

See above

Decision letter (RSOS-201898.R1)

Dear Dr Salat

On behalf of the Editors, we are pleased to inform you that your Manuscript RSOS-201898.R1 "Analysing the impact of electrification on rural attractiveness in Senegal with mobile phone data" has been accepted for publication in Royal Society Open Science subject to minor revision in

accordance with the referees' reports. Please find the referees' comments along with any feedback from the Editors below my signature.

Note that one of the reviewers expressed some concern regarding data access from Sonatel. While the journal does permit exceptions to data access requirements on grounds of confidentiality, for instance, we would ask that you make it clear to whom a data access request should be addressed - please can you ensure that a named individual and/or department, along with a contact email address, is provided in addition to the existing data access statement prior to resubmitting your manuscript?

Please submit your revised manuscript and required files (see below) no later than 7 days from today's (ie 25-Aug-2021) date. Note: the ScholarOne system will 'lock' if submission of the revision is attempted 7 or more days after the deadline. If you do not think you will be able to meet this deadline please contact the editorial office immediately.

on behalf of Professor Weisi Guo (Associate Editor) and R. Kerry Rowe (Subject Editor)
openscience@royalsociety.org

Reviewer comments to Author:

Reviewer: 1

Comments to the Author(s)

Authors have successfully addressed most of the points raised in my first report. However, in my opinion the point concerning the discussion on the exponents obtained for the gravity model has not been properly addressed. The answer of the authors seems to imply that the exponents obtained depend on the scale of the variables, and this should not be the case. In a gravity model the value as described by Eq. (3.1) the value of the constant k certainly depends on scale for the distance but the exponent γ should not. What (3.1) implies is that if, for instance, the distance is duplicated, then T decreases by a factor 2^{γ} no matter the scale in which distance has been measured (m, Km, as a fraction of the country size,...). Similarly exponents α_i and β_i should not depend on the units used to measure O_i and D_i , while the constant k does. This is why for a gravity law a discussion on the values of the exponents is relevant.

Therefore I consider that the text that authors have added at the end of 3 (b) (i) is completely misleading, besides the fact that has been added in subsection (i), before the introduction of the gravity law in subsection (ii).

As noted in my first report, I think the authors should include a discussion on the values of the exponents. For instance, the exponent of the distance, γ , is quite similar for cases shown in table 2 and relatively close to 1. However α and β are quite different, for instance for Pop/Edu, the exponent of education, β is over 6 while for Pop/Ele β less than 0.6. I suggest the authors to discuss the values of the different exponents and in what sense these values can be considered as reflecting a physical mechanism for mobility. For instance, to which extend it make sense to consider the level of education to power 6?

Once this point has been addressed, in my opinion the manuscript can be accepted for publication.

Reviewer: 2

Comments to the Author(s)

This is a solid piece of paper, using CDR data in order to help understand the impact of electricity access in developing countries. The methods are relatively standard, based on gravity models, and appear to be correct.

===PREPARING YOUR MANUSCRIPT===

===PREPARING YOUR REVISION IN SCHOLARONE===

Author's Response to Decision Letter for (RSOS-201898.R1)

See Appendix B.

Decision letter (RSOS-201898.R2)

Dear Dr Salat,

I am pleased to inform you that your manuscript entitled "Analysing the impact of electrification on rural attractiveness in Senegal with mobile phone data" is now accepted for publication in Royal Society Open Science.

on behalf of Professor Weisi Guo (Associate Editor) and R. Kerry Rowe (Subject Editor)
openscience@royalsociety.org

Appendix A

Point-by-point response to referees and Editors

We would like to thank **Reviewer #1** for their useful comments.

Why diesel engines are considered to be unreliable electric sources while solar systems are considered reliable? Is this because by diesel engines refer only to very small generators for a single household which may not be properly maintained? Also in what sense photovoltaic installations which are subject to weather variability are considered fully reliable? Is this because only large photovoltaic installations backed by storage systems (batteries) are considered?.

In the context of the paper, diesel engines refer to small individual units that are shared by a varying (small) number of households and often only work part of the day. PV is not considered "fully reliable", but only stable. This is because Senegal benefits from high solar irradiation across the country and because photo-voltaic installations require very low maintenance compared to diesel generators. We have added this explanation in the text (§2.(b)), together with references to the literature upon which these statements are based (SolarPower Europe's Senegalese PV market analysis and a World Bank report).

In the definition of the electrification rate, it is not clear what is understood by "individuals with a stable access", does this includes the individuals with households connected to "reliable sources" or does it includes also diesel engines?. Either way, authors built the index on the basis of a binary option (an individual has "stable access" or nothing), so it does not takes into account the existence of different reliability levels on the electrical access. A gradual index seems to me more appropriate, for instance assign 1 to individuals with access to reliable sources, 0.5 to those with access to less reliable sources and 0 to those with no access, and then take an average over the population.

The phrasing has been changed (§2.(b)) to better reflect that a stable access should be understood as an access to the main grid or PV installations. In the context of the paper, as defined at the beginning of the abstract and introduction, we are only interested in longer term reliable development and do not wish to include individual 'stopgap' solutions. In particular, the effective hours during which diesel engines are turned on varies widely and is not properly quantified, and therefore would not be well represented by 0.5.

In the first paragraph of 3(a), does the call distribution used to calculate the entropy for the social diversity index includes both incoming and outgoing calls or only outgoing?

The call distribution includes both incoming and outgoing calls. This information has been added to §3.(a).

In the paragraph before table 1, authors indicate that the last two columns of the table use a recomputation of the seven indices over the sub-network containing only rural nodes. However it is not explained how the recomputation is performed. For instance if in the full network A is a urban

node and B and C are rural nodes, with calls from B to A and from A to C but no calls from B to C, the sub-network does have a link between B to C or not?

The rural sub-network only retains direct links between rural nodes. The text in the paragraph before table 1 (§3.(a)) has been changed to mention this explicitly.

I suggest that at the beginning of the paragraph before table 1, "Spearman correlations" are referred as "Spearman rank correlations".

This has been updated (§3.(a)) according to the suggestion.

In table 1 authors distinguish between direct and partial correlations, a distinction which is not evident for non-statisticians. Besides table 1, partial correlations are extensively used in the rest of section 3(a) as well as in 3(b). Therefore I suggest the authors to briefly explain, either here or better in sect. 2 materials and methods, what is meant by a partial correlation between two variables controlled by a third variable and how this is evaluated using Spearman rank correlations.

The text has been rearranged and expanded to define more precisely partial correlations (§3.(a)).

In the paragraph before table 1, it is not clear what the authors mean by of the "the electrification rate variable is highly correlated to our combined socio-economic indicator". Up to this point, authors have defined 6 indexes but there is no definition of a "combined socio-economic indicator". The "combined socio-economic indicator" is also used later in section 3 (b) (i), however neither there seems to be clearly defined. Is this indicator the sum of the level of education, the employment and the quality of accommodation?. If so, shouldn't the sum be weighted since education, according to fig. 4b ranges from 6 to 15 while the values of the quality of accommodation, according to fig. S2, do not go beyond 3.1. A precise definition of the combined socio-economic indicator should be given in the main text.

The definition of the "combined socio-economic indicator" is provided at the end of paragraph 2(c), after the definitions of the other three socio-economic indicators. It is the first component of a principal component analysis on the other three indicators. We have added "defined above" in §3.(a) as a reminder, as we acknowledge that the short definition could be easily missed and have clarified the definition in §2.(c).

As authors note in the paragraph after table 1, social diversity direct correlation with electrification is close zero in rural areas, however there is no discussion on this. I suggest to add a discussion on possible causes.

We have added "These values could rather result from the heavy constraints on mobility in rural areas not allowing to develop and maintain a diverse network beyond near geographical proximity" (§ 3.(a)). This further echoes the importance of constraints on mobility underlined by reviewer #1 below in reference to the gravitation modelling.

Spearman rank correlation values in between 0.2 and 0.4 are typically considered weak and those below 0.2 very weak. All partial correlations shown in table 1 and in Fig. 2 are below 0.4. I suggest authors to include this in the discussion and, if appropriate, put it in the context of the methodology used to evaluate the partial correlations.

Indeed, considering the amount of potential interference impacting social networking patterns and decisions, a correlation of 0.2 should remain noteworthy in our context, especially on such relatively large samples. This is emphasised by the p-value < 0.05 associated to some of these lower correlations. We have changed "significant" to "statistically significant" and have elaborated more on the context of partial correlations in the paragraph under table 1 (§ 3.(a)).

In fig. 3 in section 3(b) (I) authors plot a straight lines on top of the data with an indication of the slope. However the data clearly does not have a linear behavior: It has a lower tail for low levels of electrification and an raising on for high levels of electrification typically associated to urban cells. In the text describing this figure, authors indicate high levels of correlation but the value of the correlation is not given and it is not clear that they refer to Spearman rank correlation or to Pearson linear correlation (used later for fig. 4). I think the lines associated to linear correlation should not be plotted and instead rely on Spearman rank correlation which is clearly more adequate.

The black lines are only meant to quantify the upward trend so that it becomes directly comparable with other cases. They are not meant as 'models' for the data. As indicated in the legend of fig. 3, all the p-values for the fitting of these lines are all smaller than $2.2e-16$, which guarantees that they are relevant for that purpose. We have now added in the legend that they are "(used only to quantify the upward trend)". Please note that the text relies on Spearman rank correlations, as suggested, and not on linear fittings (now made more explicit § 3.(b)(i)).

In section 3 (b) (i) fig. 4, authors make use of the Pearson correlation between the frequency of visits and the electrification rate. Why Pearson correlation is used here instead of Spearman rank correlation?. Authors should justify why one should expect a linear relation between the quantities considered here, for instance by plotting, either in the text or the supplementary material, the frequency of visits versus the electrification rate.

Fig. 4 has been updated with Spearman rank correlations, in line with the rest of the article. These correlations have higher values, although they rank among themselves similarly to the Pearson correlations. The text has been changed accordingly.

In section 3 (b) (ii), authors consider several possible variables for the gravitation models, and the results for the adjusted parameters are shown in Table 2. Authors also include a discussion on the suitability of the different parameters which is fine. However, there is no much discussion on the values of the exponents. For instance, the exponent of the distance, γ , is quite similar for cases shown in table 2 and relatively close to 1. However α and β are quite different, for instance for Pop/Edu, the exponent of education, β is over 6 while for Pop/Ele β less than 0.6. I suggest the authors to discuss the values of the different exponents and in what sense these values can be considered as reflecting a physical mechanism for mobility. For instance, to which extent it make sense to consider the level of education to power 6?.

The different values for the exponents result in part from the fact that they dwell inside different scales and therefore lose their direct comparability. We could, in theory, normalise any index (population count, population density, distance, etc.), so that it will stay within [0,1] to restore this comparability, however we believe that 'kms' and 'number of individuals' are more universal and meaningful units than 'distance compared to the diameter of the Country' (e.g.). We made the choice of focusing on comparing the variables specifically in their predictive capacity in preference to attempting to build rescaled variables for direct comparison. We have added a note about this in §3.(b)(i).

In the same section, for the case of considering the rural sub-network, I think that authors should discuss why the agreement is worst ($r^2=0.65$) than for the full network and to which extent this may imply that the mechanisms for mobility in urban and rural areas are different. Also the destination exponents β (0.18 and 0.1) are more close to zero than to 1, does this implies that the population and electrification of the destination play a little role in the mobility?. Besides, what are the values for α_1 and α_2 ?

Due to of origin constraints, there are no α_1 and α_2 values in this case. This is quite technical and explained in details in ref [37] indicated in the paragraph. We have developed the discussion provided in the next paragraph to strengthen the differences between the mechanisms for mobility between urban and rural areas (§ 3.(b)(ii)).

We would like to thank **Reviewer #2** for their positive comments.

As minor comments, I would recommend to adapt some of the choices of words, hinting that the analysis shows causal relations, and simply correlations.

We have slightly altered the wording in the abstract, last paragraph of the introduction, introduction of the result and discussion section, paragraph below table 1, conclusion of the result and discussion section and general conclusion to ensure this is reflected accurately.

Also, it appears from the map that most cities are located at the boundaries of the country, and I wondered if distance from boundary could be correlated to centrality measures, hence having an effect on the analysis.

This is an interesting idea for future work. The situation is more complex than it appears as boundaries are divided between land, sea and the specific status of Gambia that lies in the middle of the country. In addition, the region coloured in blue in figure 1(b) can be seen as a vast mixed region driven by the axis between the capital city, Dakar (western most point), and the great Mosque of Touba (eastern most part of the blue cluster).

Miscellaneous

We have added a relevant reference in the introduction that got published since the paper was submitted.

We have now made the data repository public and updated the content with an additional layer of privacy that does not impact the reproducibility of the results (slight alteration of the localisation of the antennas for privacy reasons and pre-aggregation of the mobile phone data for the entire year, as the monthly data was never used in the article).

We have added one person to the acknowledgement section.

Appendix B

London, UK,
August 27th, 2021

To: *Royal Society Open Science* Editorial Office

Dear Editors,

Please find below the point-by-point response to the comments of the Editors and referees.

> Editors

> Note that one of the reviewers expressed some concern regarding data access from Sonatel. While the journal does permit exceptions to data access requirements on grounds of confidentiality, for instance, we would ask that you make it clear to whom a data access request should be addressed - please can you ensure that a named individual and/or department, along with a contact email address, is provided in addition to the existing data access statement prior to resubmitting your manuscript?

A department within Sonatel and a contact email has been added to the data statement.

> Reviewer: 1

> Authors have successfully addressed most of the points raised in my first report. However, in my opinion the point concerning the discussion on the exponents obtained for the gravity model has not been properly addressed. The answer of the authors seems to imply that the exponents obtained depend on the scale of the variables, and this should not be the case. In a gravity model the value as described by Eq. (3.1) the value of the constant k certainly depends on scale for the distance but the exponent γ should not. What (3.1) implies is that If, for instance, the distance is duplicated, then T decreases by a factor 2^γ no matter the scale in which distance has been measured (m, Km, as a fraction of the country size,...). Similarly exponents α_i and β_i should not depend on the units used to measure O_i and D_i , while the constant k does. This is why for a gravity law a discussion on the values of the exponents is relevant. Therefore I consider that the text that authors have added at the end of 3 (b) (i) is completely misleading, besides the fact that has been added in subsection (i), before the introduction of the gravity law in subsection (ii). As noted in my first report, I think the authors should include a discussion on the values of the exponents. For instance, the exponent of the distance, γ , is quite similar for cases shown in table 2 and relatively close to 1. However α and β are quite different, for instance for Pop/Edu, the exponent of education, β is over 6 while for Pop/Ele β less than 0.6. I suggest the authors to discuss the values of the different exponents and in what sense these values can be considered as reflecting a physical mechanism for mobility. For instance, to which extend it make sense to consider the level of education to power 6?.

The previously added text has been removed and a much lengthier explanation following the suggestions made by the reviewer has been added.

> Reviewer: 2

No action required.